# Zooanthroponotic transmission of SARS-CoV-2 and host-specific viral mutations revealed by genome-wide phylogenetic analysis

Sana Naderi[1], Peter E Chen[1,2], Carmen Lia Murall[1,3], Raphael Poujol[4], Susanne Kraemer[1], Bradley S Pickering[5,6,7], Selena M Sagan[1,8]*, B Jesse Shapiro[1,9,10]*

[1]Department of Microbiology & Immunology, McGill University, Montreal, Canada; [2]Département de sciences biologiques, Université de Montréal, Montreal, Canada; [3]Public Health Agency of Canada, Winnipeg, Canada; [4]Research Centre, Montreal Heart Institute, Montreal, Canada; [5]National Centre for Foreign Animal Disease, Canadian Food Inspection Agency, Winnipeg, Canada; [6]Department of Veterinary Microbiology and Preventative Medicine, College of Veterinary Medicine, Iowa State University, Ames, United States; [7]Department of Medical Microbiology and Infectious Diseases, University of Manitoba, Winnipeg, Canada; [8]Department of Biochemistry, McGill University, Montreal, Canada; [9]McGill Genome Centre, Montreal, Canada; [10]McGill Centre for Microbiome Research, Montreal, Canada

*For correspondence:
selena.sagan@mcgill.ca (SMS);
jesse.shapiro@mcgill.ca (BJS)

Competing interest: The authors declare that no competing interests exist.

**Abstract** Severe Acute Respiratory Syndrome Coronavirus 2 (SARS-CoV-2) is a generalist virus, infecting and evolving in numerous mammals, including captive and companion animals, free-ranging wildlife, and humans. Transmission among non-human species poses a risk for the establishment of SARS-CoV-2 reservoirs, makes eradication difficult, and provides the virus with opportunities for new evolutionary trajectories, including the selection of adaptive mutations and the emergence of new variant lineages. Here, we use publicly available viral genome sequences and phylogenetic analysis to systematically investigate the transmission of SARS-CoV-2 between human and non-human species and to identify mutations associated with each species. We found the highest frequency of animal-to-human transmission from mink, compared with lower transmission from other sampled species (cat, dog, and deer). Although inferred transmission events could be limited by sampling biases, our results provide a useful baseline for further studies. Using genome-wide association studies, no single nucleotide variants (SNVs) were significantly associated with cats and dogs, potentially due to small sample sizes. However, we identified three SNVs statistically associated with mink and 26 with deer. Of these SNVs, ~⅔ were plausibly introduced into these animal species from local human populations, while the remaining ~⅓ were more likely derived in animal populations and are thus top candidates for experimental studies of species-specific adaptation. Together, our results highlight the importance of studying animal-associated SARS-CoV-2 mutations to assess their potential impact on human and animal health.

## Editor's evaluation

The authors have rigorously examined the existence of mutations that are associated with SARS-CoV-2 transmission between non-humans and human animals. The findings are important for contemporary conversations around viral niche breadth and disease emergence. The evidence

provided is very convincing, and the authors have creatively demonstrated how these sorts of GWA studies can be used to identify the genomic signature of niche breadth.

## Introduction

Coronaviruses can have broad animal host ranges, and SARS-CoV-2 is no exception. Although the animal reservoir of ancestral SARS-CoV-2 remains unknown, SARS-CoV-2 has close relatives in bats and an ancestral variant likely spilled over into humans via an intermediate animal host in a seafood market in Wuhan, China (*Worobey et al., 2022*). Although SARS-CoV-2-related coronaviruses from animals in the Wuhan market were not sampled, there have been several subsequent reports of SARS-CoV-2 transmission ('spillback') from humans to animals including in farmed mink (*Oude Munnink et al., 2021*) and wild white-tailed deer (*Odocoileus virginianus*) (*Kuchipudi et al., 2022*; *Kotwa et al., 2022*). Consequently, transmission among potential animal reservoirs is a key feature of the past and future evolution of coronaviruses. In addition to making SARS-CoV-2 elimination highly unlikely, evolution in animal reservoirs could transiently increase evolutionary rates (*Porter et al., 2023*) and potentially select for novel mutations with effects on transmission and virulence in humans (*Otto et al., 2021*). Viral adaptation to one host species might result in the tradeoff of reduced transmission in other species. For example, the Middle East respiratory syndrome coronavirus appears primarily adapted to camels, with relatively short-lived transmission chains in humans (*Dudas et al., 2018*). Alternatively, evolution in different species could potentially create new paths to peaks in the human-adaptive fitness landscape. It is speculated that the SARS-CoV-2 Omicron variant of concern (VOC) might have evolved in a non-human animal (possibly a rodent) before transmitting widely among humans (*Wei et al., 2021*). This scenario remains hypothetical and is not exclusive of other hypotheses, such as evolution in unsampled human populations, and/or in immunocompromised/chronically infected individuals. Nevertheless, ongoing transmission among humans and non-human animals and its implications for viral evolution and host adaptation deserves further study.

There have been several reports of SARS-CoV-2 transmission from humans to other animal species, and in some cases back to humans. In addition to farmed mink (*Oude Munnink et al., 2021*), there have also been reports of transmission from pet hamsters (*Yen et al., 2022*) and possible transmission events from white-tailed deer to humans in North America (*Pickering et al., 2022*). Certain mutations may improve the replication fitness of SARS-CoV-2 in animal hosts. For instance, experimental infections of ferrets (*Mustela furo*) identified a Y453F spike (S) mutation that increases viral binding to the ACE2 receptor in these animals, while decreasing infection of human airway cells (*Zhou et al., 2022*). Another potentially mustelid-adapted mutation (S:N501T) was also found to be prevalent in infected mink sampled in the United States (*Cai and Cai, 2021*; *Eckstrand et al., 2021*). Yet this mutation appears not to suffer a fitness tradeoff in humans; rather, it caused stronger binding to human ACE2 in an in vitro screen (*Starr et al., 2020*).

While such case studies have been valuable in highlighting patterns of transmission and adaptation across individual animal species since the beginning of the pandemic, a standardized, global analysis of the available data is lacking. In this study, we comprehensively compared SARS-CoV-2 sequences derived from animal hosts using a publicly available dataset. Using phylogenetic methods, we inferred transmission events between humans and four other well-sampled animals and quantified their relative frequencies. Compared to earlier studies which tended to focus on one species at a time and much smaller datasets we use an automated approach to avoid manual browsing of the tree. This not only increases the efficiency of the analysis but also reduces the potential for human error. Second, rather than simply identifying all branches involving a transition from one host species to another, our method scans the tree for such branches and retains only the most basal nodes (closest to the root) on the descendant end of a given branch. This approach overcomes the possible overcounting of nested clades as separate transmission events and is conservative in reporting possible cross-species transmission events.

Our analysis revealed a relatively high number of mink-to-human transmission events, while instances of animal-to-human transmission from cats (*Felis catus domesticus*), dogs (*Canis lupus familiaris*), or deer (*Odocoileus virginianus*) were rare. Using genome-wide association studies (GWAS), we identified mutations associated with specific animal species. We recovered the S:N501T mutation previously associated with mink, along with two other amino acid changes in other SARS-CoV-2

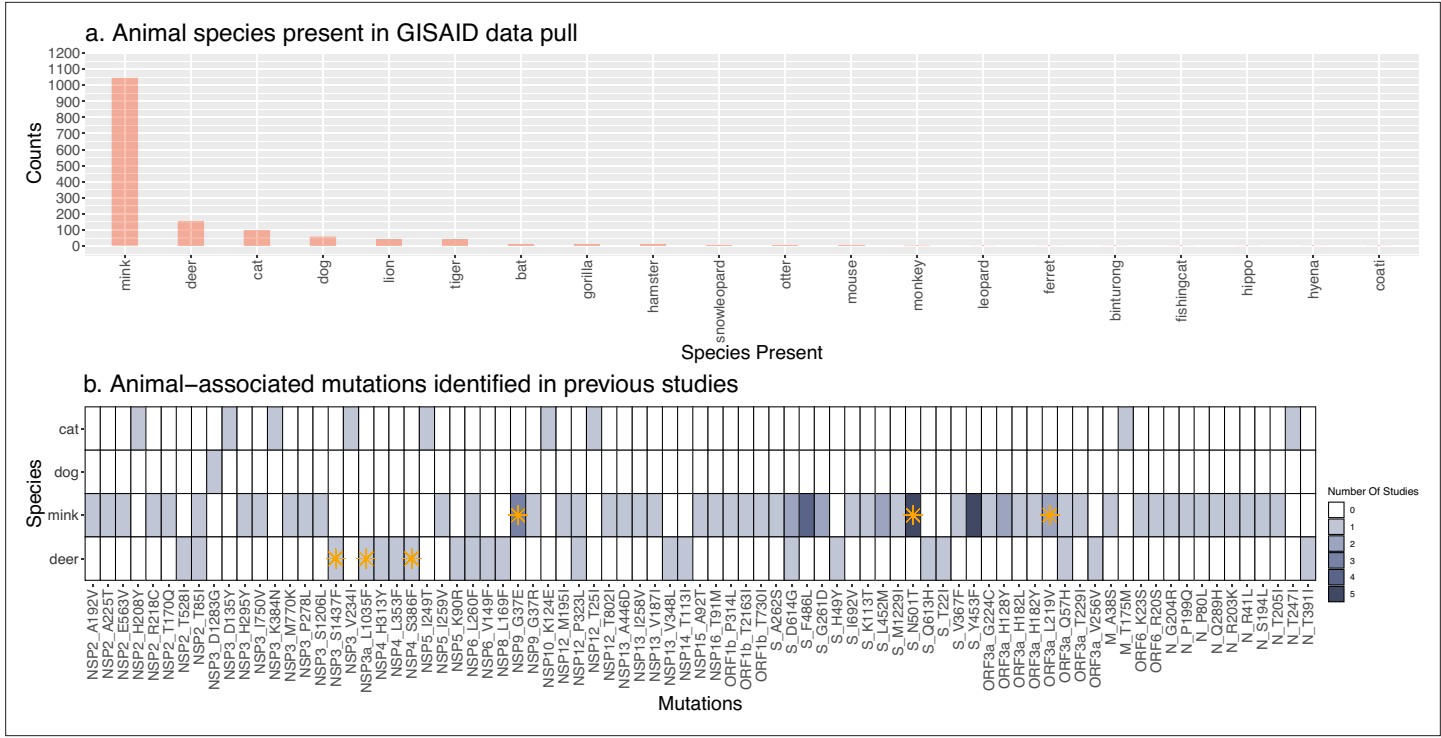

**Figure 1.** Overview of available Severe Acute Respiratory Syndrome Coronavirus 2 (SARS-CoV-2) genome sequences from different animal species. (**A**) Barplot of the number of genome sequences available in GISAID (on February 28, 2022) sampled from each animal species. Only species with 50 or more sequences were included in the current study: cat, dog, mink, and deer. (**B**) Heatmap of animal-associated mutations identified in previous publications. Darker colors indicate mutations found in a greater number of studies. Each row corresponds to one of the species included in this study, and columns correspond to mutations along the SARS-CoV-2 reference genome. Mutations identified as family-wise significant (p<0.05) in our genome-wide association studies are indicated with an orange asterisk. A detailed list of publications reporting these mutations is in ***Supplementary file 8***. Only single nucleotide variants, not insertions or deletions, are included. The heatmap illustrates the results of several key prior studies but does not represent a comprehensive meta-analysis.

genes. We also identified several novel mutations associated with deer, including both synonymous and nonsynonymous substitutions. Together, our work provides a quantitative framework for tracking SARS-CoV-2 transmission across animals from available genomic sequences and points to several candidate animal-adaptive mutations for experimental follow-up.

## Results

Although secondary spillover events of SARS-CoV-2 transmission from animals back to humans have been reported, their frequency has yet to be quantified and compared across animal species. To this end, we retrieved all available SARS-CoV-2 genome sequences sampled from non-human animals from GISAID (***Shu and McCauley, 2017***; ***Figure 1a***, ***Supplementary file 1***). After applying sequence quality filters (***Supplementary file 2***) and considering only animals with 50 or more sequences, we were left with four species: cat, dog, mink, and deer. Our filters excluded common experimental animals such as mice and ferrets (***Figure 1a***). For each of the four animal species, we extracted a similar number of closely-related human-derived sequences (Methods). If these closely-related animal-human pairs represent recent transmission chains, we would expect them to come from the same geographic region. Consistent with this expectation, we found that 95.4% of deer-derived sequences share the same sampling location as their closest human-derived relatives, and this percentage is slightly lower for cats (85.6%), dog (89%), and mink (91.7%). The high percentage of deer could be due to higher sampling efforts in North America, the origin of all deer sequences. More generally, the variation in these percentages could be explained by other sampling biases. For example, cats and dogs might be undersampled relative to mink and deer.

To investigate potential animal-to-human transmission events in greater detail, including the direction of transmission, we analyzed viral phylogenetic trees. For each species, the animal-derived sequences and their closest relative human-derived sequence were combined with ten random sub-samplings of human-derived sequences (n ≈ 50 per subtree per month of the pandemic), from which we inferred ten replicate phylogenies per species, providing an assessment of phylogenetic uncertainty. Using ancestral state reconstruction as previously described (*Murall et al., 2021*), we counted the most basal animal-to-human (*Figure 2a–d*) and human-to-animal transitions on each tree (*Supplementary file 3*). Representative detailed trees are available in *Figure 2—source data 1–4*. Ancestral state reconstruction can be biased by differential sampling across species. Such bias tends to be more severe close to the root of the tree (*De Maio et al., 2015*) but, in our case, the root is known to be human (as we excluded more divergent animal outgroups). Most of the inferred transmission events are close to the tips and far from the root (*Figure 2*), which we expect to minimize (but not entirely eliminate) bias. Our goal is, therefore, not to infer absolute rates of cross-species transmission, but rather to provide a consistent comparative framework for interspecies transmission.

To determine a more conservative lower bound for the transmission counts, we excluded transition branches with <75% bootstrap support (Methods). As further validation, we performed a permutation test to determine whether the estimated transmission counts converged on a non-random value. We performed the same ancestral state reconstruction on 1000 permutations of each of the 40 phylogenies whose tip labels (animal or human) were shuffled randomly, and the number of transmissions in both directions was recorded. This permutation test revealed that in both directions (animal-to-human and human-to-animal) the observed transmission counts in mink, deer, and cat falls within a narrow range (standard deviations of 10.32, 0.96, and 0.52, respectively) compared to the permutations (68.83, 2.85, and 1.90) while the observed counts in a dog (standard deviation of 0.95) is similar to the permutations (0.90). In the human-to-animal direction, the observed standard deviations in transmission counts are 0.48, 0.48, 2.15, and 5.2 for cat, dog, mink, and deer, respectively, compared to permuted values of 8.91, 1.68, 67.37, and 18.44. These results show that, in general, our inferences of transmission events converge upon stable values, but that the estimate for dog-to-human transmission might be less reliable.

Based on the bootstrap-filtered counts, we inferred fewer than two transmission events on average to humans from dogs and deer, and around four transmissions on average from cats to humans (*Figure 3a*, *Table 1*). In contrast, there were an average of 38 or more transmission events inferred from mink back to humans. The upper bound (bootstrap unfiltered) estimates are higher, but the pattern of much higher transmission from mink is retained. The inferred number of transmissions in the opposite direction, from humans to other animals, was generally higher and much more uniform across species (*Figure 3b*, *Table 1*), with lower bounds in the range of 31.5–58.5 events. The relatively higher sampling of sequences from humans than from other animals may have inflated the number of inferred human-to-animal events relative to animal-to-human events. Mink are also better sampled than the other animal species, and further sampling of other animals could identify more transmission events. On the other hand, the number of inferred transmission events is not clearly related to the sample size. For example, despite their smaller sample size compared to deer, cats tend to be involved in more transmission events to and from humans (*Table 1*). Despite these caveats, it is notable that human-to-animal transmission events are relatively constant across animal species, while mink-to-human transmission is much more frequently detected compared with other animals.

Next, we sought to identify mutations associated with each animal species compared to humans. These mutations could be candidates for species-specific adaptations. Dozens of mutations in the SARS-CoV-2 genome have been reported in association with different animal species (particularly mink; *Figure 1b*) but it is unclear how many of these associations are statistically significant. To test for significant associations, we conducted GWAS using POUTINE, a method that implicitly controls for viral population structure and linkage (non-independence) between mutations by considering only homoplasic mutations that are identical by state, not identical by descent, and that have occurred independently of multiple times in the SARS-CoV-2 phylogeny (*Chen and Shapiro, 2021*). We performed a separate GWAS to identify mutations associated with each species and replicated the GWAS on each of the ten replicate trees described above.

We identified numerous SNVs with high statistical significance associated with mink and deer, but none in cats or dogs (*Figure 4*). In all cases, we used a significance cutoff of a family-wise p<0.05 to

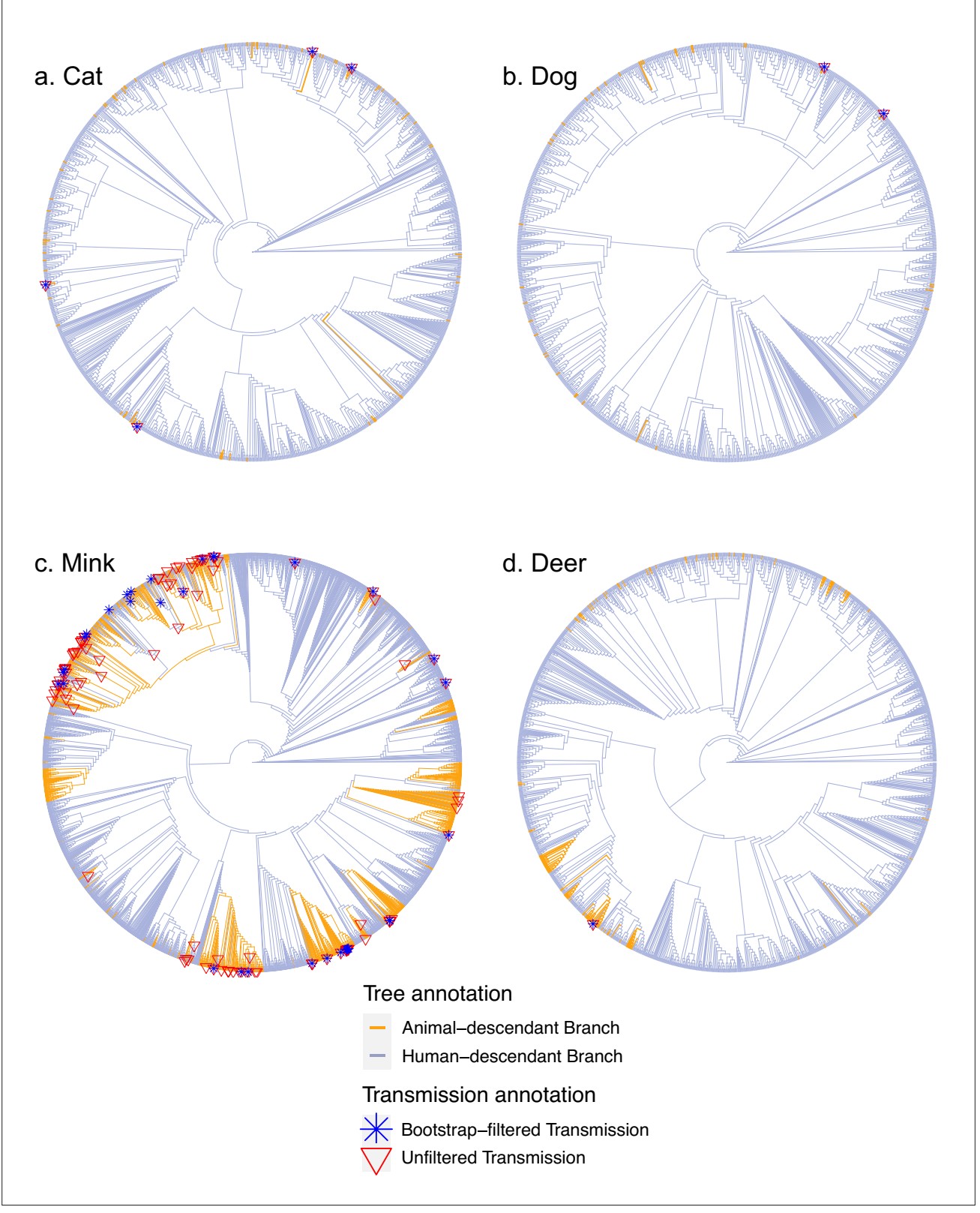

**Figure 2.** Transmission events inferred from non-human animals to humans. Panels a-d display a representative tree for every species with animal-to-human transmissions marked on the tree. More detailed versions of these trees are in . Trees are rooted with the Wuhan reference genome (from one of the first sampled human COVID-19 patients).

*Figure 2 continued on next page*

*Figure 2 continued*

The online version of this article includes the following source data for figure 2:

**Source data 1.** Detailed representative phylogeny of cat- and human-derived Severe Acute Respiratory Syndrome Coronavirus 2 (SARS-CoV-2) sequences.

**Source data 2.** Detailed representative phylogeny of dog- and human-derived Severe Acute Respiratory Syndrome Coronavirus 2 (SARS-CoV-2) sequences.

**Source data 3.** Detailed representative phylogeny of mink- and human-derived Severe Acute Respiratory Syndrome Coronavirus 2 (SARS-CoV-2) sequences.

**Source data 4.** Detailed representative phylogeny of deer- and human derived Severe Acute Respiratory Syndrome Coronavirus 2 (SARS-CoV-2) sequences.

correct for multiple hypothesis testing. The mink GWAS revealed three unique SNVs (*Table 2*). One of these hits appears in all ten replicates and the remaining two appear in at least half of the replicates. All three of these mutations are non-synonymous, including S:N501T which was previously associated with a mink outbreak in the United States (*Cai and Cai, 2021*). Inspecting the distributions of these GWAS hits across the tree reveals several independent origins. For example, S:N501T occurred independently in The Netherlands, Denmark, Latvia, Lithuania, Spain, France, and the USA (*Figure 2—source data 3*), explaining the strong association detected by POUTINE. The deer GWAS revealed a total of 26 unique statistically significant SNVs, of which seven appear in all ten replicates, and five in at least half (*Table 3*). Out of these 26 hits, five are intergenic (within the 5' and 3' UTRs)

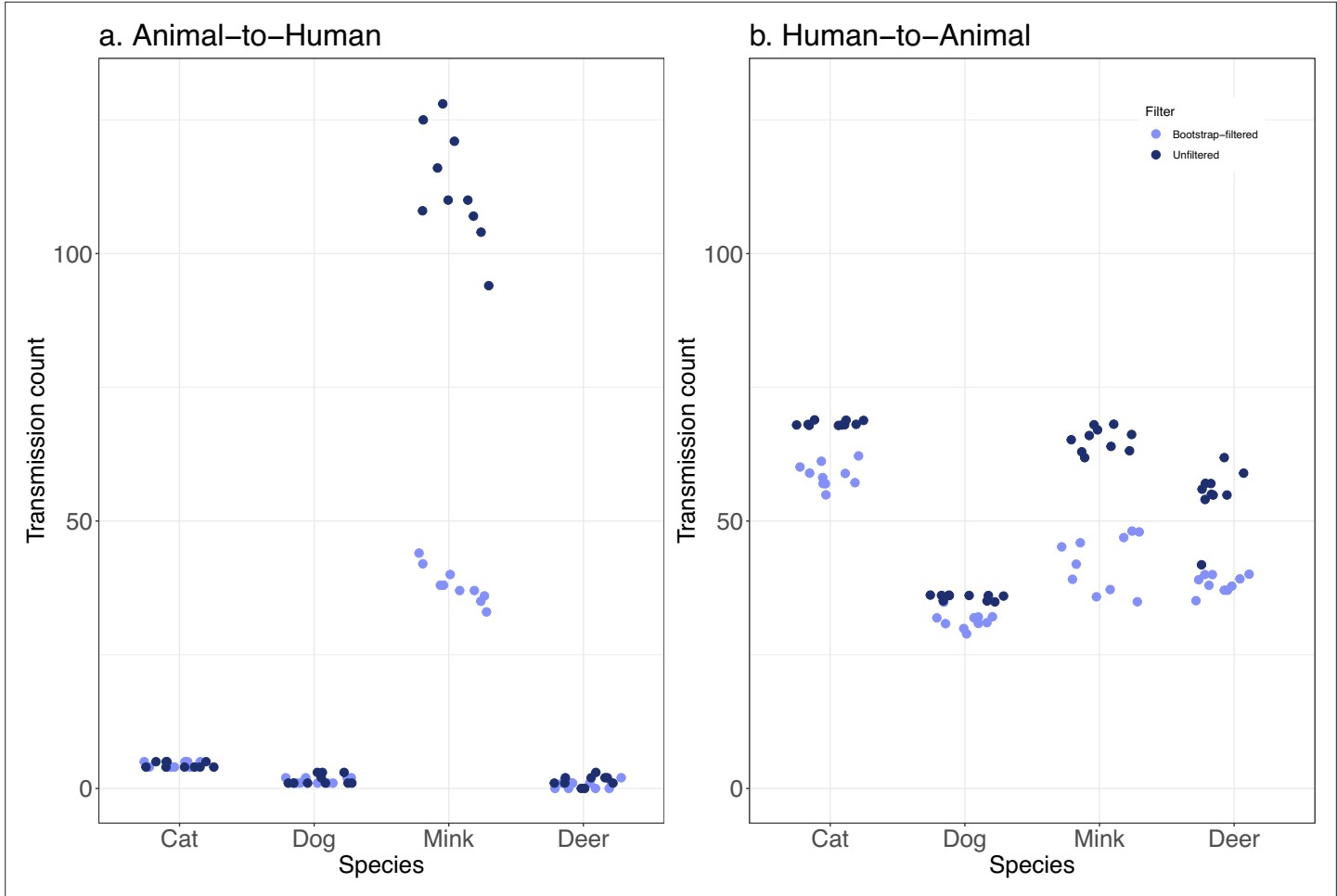

**Figure 3.** Transmission events from animals-to-humans are rarely detected, except from mink. The distribution of inferred transmission counts (across 10 replicate trees) in each animal species, in both bootstrap-filtered and unfiltered trees are shown in **A** the animal-to-human direction, and **B** the human-to-animal direction. Points are plotted with jitter to avoid overlap.

**Table 1.** Average inferred transmission events between humans and animals.

| Average inferred number of transitions (filtered – unfiltered) | Mink (n=1038) | Deer (n=134) | Cat (n=78) | Dog (n=39) |
|---|---|---|---|---|
| Animal-to-human | 38.0–112.3 | 0.7–1.4 | 4.4–4.4 | 1.4–1.7 |
| Human-to-animal | 42.3–65.2 | 38.3–55.2 | 58.5–68.3 | 31.5–35.7 |

and 12 are synonymous mutations. Notably, 21 of the hits are C>U transition mutations. The seven hits found in all ten replicates clearly occur multiple times independently in different branches of the tree; for example, ORF1ab:N4899I (which affects amino acid 507 of the mature RNA-dependent RNA polymerase protein, nsp12) occurred at least twice independently in both the states of New York and Iowa (*Figure 2—source data 4*).

We next asked whether mutations identified by GWAS plausibly occurred in an animal host or if they were more likely circulating in a local human population before being transmitted to a different animal species. While both these categories of mutations are statistically associated with a particular animal species, the former are particularly good candidates for species-specific adaptations. To address this question, we obtained the global frequency of the minor allele of each significant GWAS hit in Cov-Spectrum (*Chen et al., 2022*) – a significantly larger database than our downsampled trees used for GWAS. The minor alleles were generally rare (<1% frequency; *Supplementary files 4 and 5*), as expected for non-human animal-associated mutations in a large database dominated by human sequences. To test the hypothesis that animal-associated mutations arose not in animals but in a local human population that then transmitted to animals in the same region, for each GWAS hit we first defined the 'in' region in which the animals containing the mutation were sampled and the 'out' regions including all other regions. We then performed a Fisher's exact test to determine if human-derived sequences containing the animal-associated mutation were enriched in the 'in' region, resulting in an odds ratio (OR) significantly higher than one (Methods). For mink, two GWAS hits had OR significantly greater than one (p<0.0001, ). Only the S:N501T mutation had OR significantly less than one, making it the best candidate for having arisen in a mink host, rather than in a human who later transmitted the mutant virus to a mink. For deer, the majority of GWAS hits (18/26) could be explained by transmission from the local human population (OR >1) while the remaining eight could not (OR <1 or not significantly different than one, *Table 3*).

Finally, we asked if mutations identified as significantly associated with a specific animal by GWAS tended to occur on branches with inferred human-to-animal transitions, as would be expected for species-specific adaptive mutations. To do so, using ancestral sequence reconstruction, we identified mutations that occurred along these transition branches. As expected, all three significant mink-associated mutations occurred multiple times on human-to-mink transition branches, with the human parent node carrying the reference allele and the animal descendant node carrying the mink-associated alternate allele (*Supplementary file 6*). Of the 26 deer-associated mutations, 22 occurred multiple times on human-to-deer branches, of which 21 carry the exact nucleotide substitution identified by GWAS (*Supplementary file 7*). By contrast, none of the substitutions identified by GWAS occurred on animal-to-human branches, consistent with species-specific adaptation.

## Discussion

### Summary and caveats

SARS-CoV-2 can be considered a generalist virus, able to infect a range of different species – perhaps more so than other viruses, and even other coronaviruses such as MERS with more restricted host preferences (*Dudas et al., 2018*). Yet SARS-CoV-2 may still replicate and transmit better in certain species than others, providing opportunities for species-specific adaptive mutations to arise. SARS-CoV-2 transmission between humans and animals has typically been studied in individual species in isolation, and using heterogeneous methods and datasets. Similarly, viral mutations associated with particular species have been reported, but largely in experimental studies or region-specific sampling efforts that are difficult to generalize. Our goal in this study was to apply standardized methods to identify animal-to-human transmission events and to discover animal-associated mutations using available viral genomic data. The results are likely biased by uneven sampling efforts

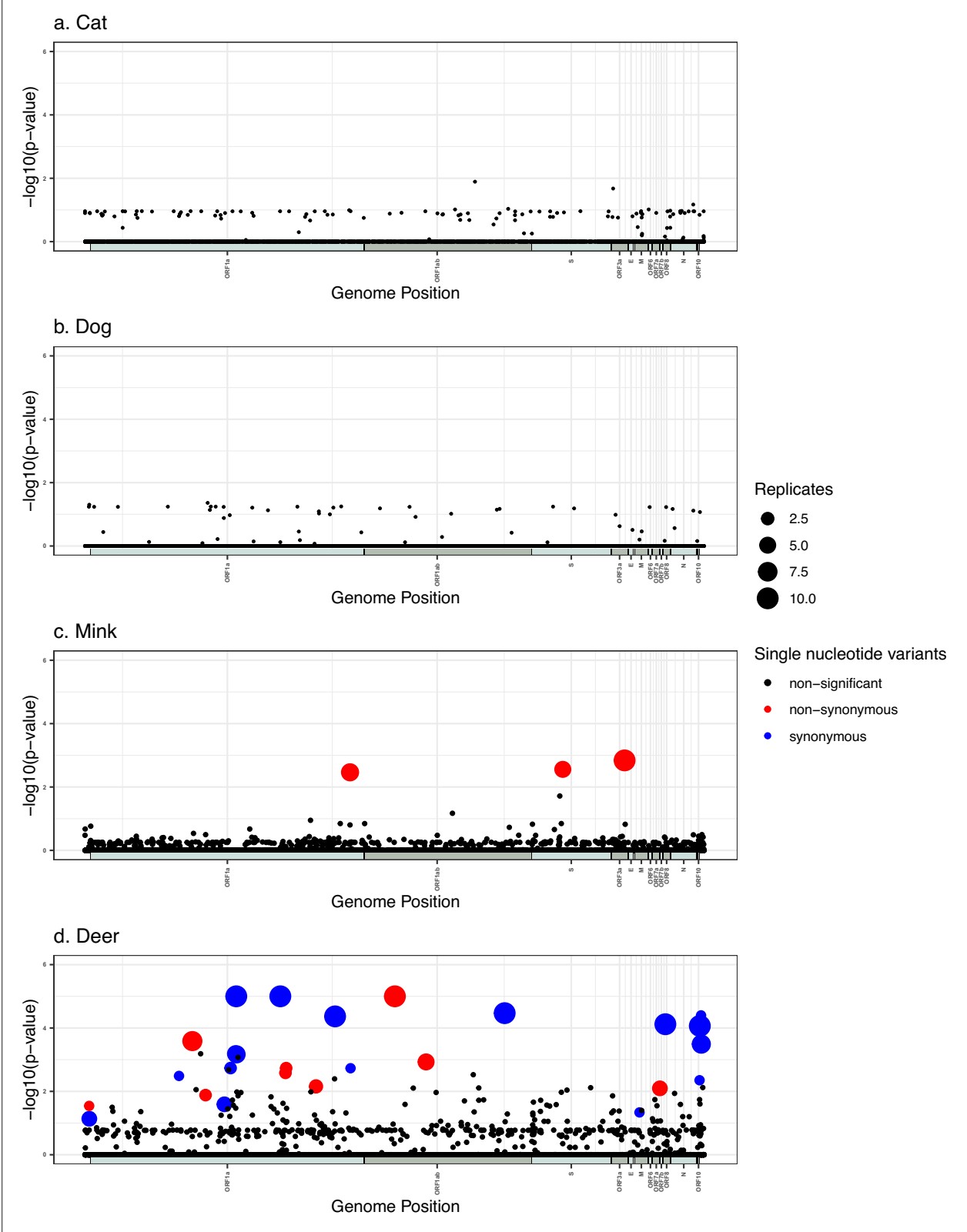

**Figure 4.** Manhattan plots summarizing genome-wide association studies (GWAS) hits in each animal species. In every panel, the x-axis represents the nucleotide position in the Severe Acute Respiratory Syndrome Coronavirus 2 (SARS-CoV-2) reference genome and the y-axis represents the -log10 of the pointwise *p*-values averaged over replicates. ORFs are shown as alternating shaded bars along the x-axis. Statistically, significant hits with family-wise corrected *p*-values of lower than 0.05 are shown in red (non-synonymous) or blue (synonymous), while non-statistically significant p-values are in black.

**Table 2.** Single-nucleotide variants associated with mink by genome-wide association studies (GWAS).

"Pos." refers to the nucleotide position in the reference genome. Homoplasy counts in focal animals (cases), humans (controls), and p-values are averaged across replicates in which the site's family-wise p-values were <0.05. Where applicable, amino acid positions refer to the polyprotein with mature protein positions in parenthesis. The 'local transmission odds ratio' is the result of a Fisher's exact test of the likelihood that the alternate base (animal-associated minor allele) was enriched in the local human population where the mink sequences bearing the alternate base were sampled (Methods). n.s., not significant. Odds ratio p-value: *<0.05, **<0.01, ***<0.001.

| Pos. | Ref. base | Alt. base | Amino acid change | Gene | Homoplasy count in focal animal | Homoplasy count in humans | p-value (pointwise) | p-value (familywise) | Significant in N replicates | Local transmission odds ratio |
|------|-----------|-----------|-------------------|------|----------------------------------|---------------------------|---------------------|----------------------|------------------------------|-------------------------------|
| 26047 | U | G | L219V | ORF3a | 6 | 0 | 0.0014 | 0.0365 | 10 | 3.93 *** |
| 12795 | G | A | G4177E (nsp9 G37E) | ORF1ab/ pp1ab/nsp9/ replicase | 6 | 0 | 0.0015 | 0.0368 | 6 | 7.53 *** |
| 23064 | A | C | N501T | Spike/S1/RBD/ binds ACE2 | 6.4 | 0 | 0.0010 | 0.0258 | 5 | 0.48 *** |

**Table 3.** Single-nucleotide variants associated with deer by genome-wide association studies (GWAS).
"Pos" refers to the nucleotide position in the reference genome. Homoplasy counts in focal animals (cases), humans (controls), and p-values are averaged across replicates in which the site's family-wise p-values were <0.05. Where applicable, amino acid positions refer to the polyprotein with mature protein positions in parenthesis. IG, Intergenic. The 'local transmission odds ratio' is the result of a Fisher's exact test of the likelihood that the alternate base (animal-associated minor allele) was enriched in the local human population where the deer sequences bearing the alternate base were sampled (Methods). n.s., not significant. Odds ratio p-value: *<0.05, **<0.01, ***<0.001.

| Pos. | Ref. base | Alt. base | Amino acid change | Gene | Homoplasy count in focal animal | Homoplasy count in humans | p-value (pointwise) | p-value (familywise) | Significant in N replicates | Local transmission odds ratio |
|---|---|---|---|---|---|---|---|---|---|---|
| 7303 | C | U | I2346I (nsp3 I1524I) | ORF1a/pp1ab/pp1a/nsp3 | 17.8 | 1.2 | 9.99E-06 | 9.99E-06 | 10 | 2.51*** |
| 9430 | C | U | I3055I (nsp4 I292I) | ORF1a/pp1ab/pp1a/nsp4 | 15.2 | 6.2 | 9.99E-06 | 9.99E-06 | 10 | 2.20*** |
| 14960 | A | U | N4899I (nsp12 N507I) | ORF1ab/pp1ab/nsp12/RdRp | 7.8 | 0.1 | 9.99E-06 | 1.09E-05 | 10 | 0** |
| 20259 | C | U | F6665F (nsp15 F213F) | ORF1ab/pp1ab/nsp15 | 4.8 | 0.1 | 3.39E-05 | 0.0013 | 10 | n.s. |
| 28016 | C | U | F41F | ORF8 | 4 | 0 | 7.59E-05 | 0.0061 | 10 | 6.09*** |
| 12073 | C | U | D3936D (nsp7 D67D) | ORF1a/pp1ab/pp1a/nsp7 | 5.2 | 1.1 | 4.29E-05 | 0.0025 | 10 | n.s. |
| 29679 | C | U | IG | 3'UTR | 5 | 1.8 | 8.59E-05 | 0.0055 | 10 | 3.17*** |
| 5184 | C | U | P1640L (nsp3 P822L) | ORF1a/pp1ab/pp1a/nsp3 | 4.6 | 1.6 | 0.0002 | 0.0115 | 8 | 2.61*** |
| 29750 | C | U | IG | 3'UTR/S2M | 5 | 2.6 | 0.0002 | 0.0103 | 7 | 3.12*** |
| 7318 | C | U | F2351F (nsp3 F1533F) | ORF1a/pp1ab/pp1a/nsp3 | 4 | 0.3 | 0.0001 | 0.0114 | 6 | 3.80*** |
| 16466 | C | U | P5401L (nsp13 P77L) | ORF1ab/pp1ab/nsp13/Hel | 5 | 1 | 4.99E-05 | 0.0019 | 5 | 4.09*** |

*Table 3 continued on next page*

Table 3 continued

| Pos. | Ref. base | Alt. base | Amino acid change | Gene | Homoplasy count in focal animal | Homoplasy count in humans | p-value (pointwise) | p-value (familywise) | Significant in N replicates | Local transmission odds ratio |
|---|---|---|---|---|---|---|---|---|---|---|
| 7267 | C | U | F2334F (nsp3 F1516F) | ORF1a/pp1ab/pp1a/nsp3 | 4.4 | 0.8 | 9.39E-05 | 0.0079 | 5 | 2.79*** |
| 210 | G | U | IG | 5'UTR/SL5a | 4 | 0.5 | 0.0001 | 0.0136 | 4 | 3.98*** |
| 6730 | C | U | N2155N (nsp3 N1337N) | ORF1a/pp1ab/pp1a/nsp3 | 4 | 0.75 | 0.0002 | 0.0168 | 4 | 1.81** |
| 27752 | C | U | T120I | ORF7a | 4 | 0.75 | 0.0002 | 0.0169 | 4 | 4.03*** |
| 11152 | C | U | V3629V (nsp6 V60V) | ORF1a/pp1ab/pp1a/nsp6 | 4 | 0.7 | 0.0002 | 0.0153 | 3 | 0.80** |
| 5822 | C | U | L1853F (nsp3 L1035F) | ORF1a/pp1ab/pp1a/nsp3 | 4 | 0.5 | 0.0001 | 0.0118 | 2 | n.s. |
| 9711 | C | U | S3149F (nsp4 S386F) | ORF1a/pp1ab/pp1a/nsp4 | 4 | 0.5 | 8.49E-05 | 0.0118 | 2 | 0.56** |
| 9679 | C | U | F3138F (nsp4 F375F) | ORF1a/pp1ab/pp1a/nsp4 | 4 | 0 | 9.49E-05 | 0.0067 | 2 | 2.32*** |
| 7029 | C | U | S2255F (nsp3 S1437F) | ORF1a/pp1ab/pp1a/nsp3 | 4 | 0.5 | 0.0002 | 0.0149 | 2 | 0.22*** |
| 29738 | C | A | IG | 3'UTR/S2M | 4 | 0 | 3.99E-05 | 0.0059 | 1 | n.s. |
| 26767 | U | C | I82T | ORF5/M | 4 | 0 | 8.99E-05 | 0.0057 | 1 | 4.09*** |
| 203 | C | U | IG | 5'UTR/SL5a | 4 | 1 | 0.0003 | 0.0191 | 1 | 5.94*** |
| 12820 | A | G | L4185L (nsp9 L45L) | ORF1a/pp1ab/pp1a/nsp9 | 5 | 1 | 3.99E-05 | 0.0009 | 1 | 4.52*** |
| 4540 | C | U | Y1425Y (nsp3 Y607Y) | ORF1a/pp1ab/pp1a/nsp3 | 4 | 1 | 0.0002 | 0.0239 | 1 | 2.80*** |
| 29666 | C | U | L37F | ORF10 | 4 | 1 | 0.0002 | 0.0219 | 1 | 1.54*** |

across species. Specifically, oversampling of human-derived sequences could bias the ancestral state reconstruction toward human-to-animal rather than animal-to-human transmission. In the future, such biases might be accounted for using a Bayesian structured coalescent approximation (*De Maio et al., 2015*). However, we do not expect this bias to be severe, for three reasons. First, most of our inferred transmission events are close to the tips, while bias towards the most common state appears worst close to the root (*De Maio et al., 2015*). Second, the number of inferred transmission events involving each animal species is not clearly associated with its sample size. Third, we also used permutations to flag certain inferences (e.g. dog-to-human transmission) as unreliable in our current sample. Overall, our transmission analysis provides a 'level playing field' upon which to assess the results of previous animal-specific studies. Our genome-wide association studies considered only homoplasic muta-tions that occur multiple times independently in the SARS-CoV-2 phylogeny. This provides strong protection against false-positive associations that could occur due to population structure in clonal pathogens (*Chen and Shapiro, 2021*) at the expense of missing non-homoplasic true-positives. Like most GWAS approaches, we test each nucleotide site independently, implicitly assuming additive fitness effects. This may be a reasonable assumption, as non-additive epistatic fitness effects have been inferred to be negligible in SARS-CoV-2 evolution to date, at least in the spike receptor-binding domain (*Rochman et al., 2022*). As genome sequences continue to be sampled from different animal species, the approaches described here will allow researchers to further study the evolution and trans-mission of SARS-CoV-2 and other pathogens across animal species. Sampling more sequences from diverse animals (including the four analyzed here, as well as others) will enable more robust estimates of secondary zoonotic and zooanthroponotic transmission events and species-specific adaptations.

## What explains variation in transmission and evolution across species?

Methodological and sampling considerations aside, there are two major factors affecting rates of transmission across animal species: structural similarity in viral receptor proteins (or more general phylogenetic similarity) and rates of contact between species. These factors have been used to predict cross-species viral transmission, with habitat overlap playing a dominant role over phyloge-netic similarity at least on a global scale (*Carlson et al., 2022*). On more local scales, contact with pets such as cats and dogs will be high for a large number of pet owners, while contact with minks may be high transiently, or for a subset of the population working on farms. Similarly, contact with deer may be restricted to hunters,or may involve indirect contact with human waste. A recent study compared ACE2 orthologs across mammalian species and classified deer ACE2 as having 'high' structural simi-larity to human ACE2; meanwhile, cat similarity was 'medium,' dog was 'low,' and mink species were 'very low' (*Damas et al., 2020*). Clearly, large mink outbreaks have occurred despite a relatively dissimilar ACE2 structure – and the 'mismatch' between SARS-CoV-2 spike and mink ACE2 could impose a strong selection for mink-adapted spike mutations (discussed below). Meanwhile, if deer ACE2 was already structurally similar to human ACE2, this could explain the lack of deer-associated mutations in spike. While it is tempting to speculate that the many other deer-associated mutations outside of spike could provide deer-specific adaptations that reduce transmission back to humans, it is equally possible that the rarity of deer-to-human transmission is explained by relatively low contact rates. As SARS-CoV-2 continues to evolve and transmit in both deer and humans, ongoing sequencing efforts might allow us to test the relative importance of these hypotheses. In summary, while global-scale predictions of zoonotic risk are possible based on contact rates and phylogenetic similarity, it is more difficult to predict the precise risk for a particular species of interest (e.g. mink vs. deer), which depends on detailed life history and genetic factors. However, combining estimates of habitat overlap with the genomic analyses described here could enable more fine-grained predictions.

Consistent with previous reports (*Oude Munnink et al., 2021*), mink had the largest number of inferred transmission events to humans. This could be because mink have more opportunities to interact with humans on mink farms, whereas contact between deer and humans is more limited and potentially seasonal (*Kuchipudi et al., 2022*). However, mink outbreaks could be more frequently reported due to higher surveillance of mink farms and noticeable symptomatic diseases in these animals (*Oreshkova et al., 2020*). As for cats and dogs, we inferred much lower transmission to humans, suggesting that they might be 'dead-ends' for the virus. This does not mean that SARS-CoV-2 transmission from dogs or cats to humans is not possible, and it may be more readily detected with deeper sampling or in prospective household transmission studies. Transmission from humans

to animals was relatively uniform across species, with somewhat higher inferred transmission to cats than to other species. Overall, our results support the previous reports of relatively high rates of mink-to-human transmission, and only rare or anecdotal transmission from other animals, such as deer (*Pickering et al., 2022*).

Despite the large sample size of mink-derived viral sequences, we only detected three mink-associated SNVs using GWAS, including the previously identified S:N501T mutation. This is consistent with relatively little time for SARS-CoV-2 to adapt to mink between transmission cycles in humans. In contrast, we detected many more SNVs associated with deer despite a smaller sample size. Even if we only consider the eight SNVs less likely to have arisen in the human population before transmitting to deer, or the seven SNVs detected in all replicate GWAS runs, there are still more than twice as many deer-associated than mink-associated GWAS hits. This could suggest a greater number of deer-adapted SNVs compared to mink-adapted SNVs, perhaps due to multiple uninterrupted cycles of deer-deer transmission. Sustained deer-deer transmission is also supported by previous studies (*Kuchipudi et al., 2022*), including the observation of relatively divergent SARS-CoV-2 genomes (*Pickering et al., 2022*). Asymptomatic or mild disease in deer relative to mink (*Oreshkova et al., 2020*) might also allow more opportunity for evolution and transmission during prolonged infections. Despite pruning out highly divergent branches, including several of the Canadian sequences involved in a potential deer-to-human transmission event (*Pickering et al., 2022*), some relatively long deer-associated branches are notable in our study, suggesting rapid evolution in deer (*Figure 2d*). Together, these results point to deer as an important reservoir of novel SARS-CoV-2 mutations. Ongoing monitoring of deer-to-human transmission events is, therefore, warranted.

## Lack of GWAS hits in cats or dogs

Several viral mutations have been previously associated with cats or dogs (*Elaswad et al., 2020*), but we found no statistically significant mutations associated with either of these species (*Figure 1b*). This could be due to limited viral adaptation, which would be expected if cats and dogs are effectively dead-end hosts for the virus, with little time for cycles of intra-species transmission. Alternatively, the limited sample size for these species could have limited GWAS power to identify adaptive mutations. Given the higher viral load and shedding in cats compared to dogs (*Bosco-Lauth et al., 2020*; *Shi et al., 2020*), we expect greater adaptation and transmission in cats; however, further data will be needed to test this expectation.

## Mink GWAS hits

We identified three statistically significant mutations associated with mink. The substitution ORF3a:L219V, which appears in all ten GWAS replicates, has been previously detected as a substitution associated with mink, in the ORF3a accessory gene (*Elaswad et al., 2020*). However, the previous detection was not tested for statistical significance. Similarly, the ORF1ab:G4177E (nsp9:G37E) mutation has been identified previously in mink-derived sequences (*Eckstrand et al., 2021*). The S:N501T substitution has also been previously associated with mink (*Lu et al., 2021*; *Elaswad et al., 2020*) and ferret (*Zhou et al., 2022*). The mink-associated S:N501T substitution increases binding to human ACE2 (*Starr et al., 2020*), and the S:N501Y substitution has been detected in several human SARS-CoV-2 VOCs with higher transmissibility, such as the Alpha variant (*Liu et al., 2022*; *Vöhringer et al., 2021*). Certain other mutations that have been previously associated with mink, including S:Y453F (*Zhou et al., 2022*; *Lu et al., 2021*; *Elaswad et al., 2020*) and S:L260F (*Adney et al., 2022*) that were suggested to arise as a result of rapid adaptation are not statistically significant in our GWAS. While such mutations could be truly animal-associated and were not picked up in GWAS due to limited power or sampling, others may be anecdotal reports that do not survive the scrutiny of rigorous statistical testing, or associations identified in laboratory conditions that are not currently observed in nature.

Our GWAS approach was designed to identify species-specific mutations, which could increase fitness in one species at the expense of others. It is less likely to identify mutations that increase fitness across species. One possible example is the S:Y453F mutation, associated with mink in several studies but not in our GWAS. In all replicate trees, the mutation occurred four times as a homoplasy in mink but never or only once in humans, resulting in a pointwise *p*-value <0.05 that did not survive correction for multiple hypothesis testing (family-wise *p*>0.05). In cell culture experiments, S:Y453F

decreased infection of human airway cells (*Zhou et al., 2022*). The mutation has been observed in human-derived sequences, but they are at quite a low frequency in Cov-Spectrum. The non-significant GWAS results for this mutation could, therefore, be explained by its occurrence in mink but also (more rarely) in humans. We performed ancestral state reconstruction of this mutation on the ten replicate mink trees, and observed that S:Y453F occurred 17 times on human-to-human transition branches, 30 times on human-to-mink branches, 85 times on mink-to-mink branches, and never on mink-to-human branches. The frequent occurrence of S:Y453F on mink-to-mink and human-to-mink branches is consistent with its adaptive value in mink, but its occasional occurrence on human-to-human branches suggests it does not completely prevent transmission among humans. This and other mutations that could potentially increase host range – or at least those that increase fitness in one species without a strong tradeoff in another species – are of particular relevance but might require new or adapted GWAS methods to detect.

## Deer GWAS hits

Many of the deer-associated mutations are synonymous and occur broadly across the genome outside of the relatively well-studied spike (S) protein (*Figure 4*). These are primarily transition mutations, which are typically generated at a higher frequency than transversions. The deer-associated synonymous mutations showed no particular bias toward codons that are preferred in deer relative to humans; therefore, selection for codon usage optimization does not easily explain these associations (data not shown). Nonetheless, the vast majority of mutations were C>U transitions, which may be a reflection of APOBEC1-mediated RNA editing, which results in the deamination of cytosine to uracil in single-stranded RNA (*Harris and Dudley, 2015*; *Salter and Smith, 2018*). Consistent with this hypothesis, C>U transitions in deer contained the consensus sequence [U/A][U/A]C[A/U][A/U], which resembles that observed for human APOBEC1-mediated deamination [AU]C[AU]; however, it remains to be seen whether the deer APOBEC1 isoform has the same substrate specificity or whether there is increased APOBEC expression or activity in deer tissue (*Di Giorgio et al., 2020*; *Rosenberg et al., 2011*). Alternatively, the C>U transitions may be related to RNA secondary structure or nucleotide composition biases required for genome condensation during viral replication organelle biogenesis or virion assembly in the deer host.

Our GWAS revealed several mutations in deer that localized to distinct RNA secondary structures in the 5′ and 3′ UTRs of the viral RNA. Specifically, in the 5′ UTR, both mutations (C203U and G210U) localized to stem-oop 5 a (SL5a), a highly conserved stem-loop structure previously implicated in virion assembly in related coronaviruses (*Yang and Leibowitz, 2015*; *Morales et al., 2013*; *Miao et al., 2021*). Similarly, in the 3′ UTR, all the mutations localized to the 3′ terminal stem-loop structure, with two specifically localized to the S2M region (C29738A and C29750U) (*Yang and Leibowitz, 2015*; *Gilbert and Tengs, 2021*; *Wacker et al., 2020*). Interestingly, while the 3′ terminal stem-loop structure is known to be hypervariable, the S2M region is highly conserved in sequence and structure across a wide range of RNA viruses, including members of the *Astroviridae, Caliciviridae, Picornaviridae,* and *Coronaviridae* families (*Tengs et al., 2013*; *Gilbert and Tengs, 2021*). While the role of this S2M region is poorly understood, its strict conservation across a range of positive-strand RNA viruses suggests that it may play an important role in the viral life cycle. Notably, both the mutations identified by GWAS are predicted to maintain the overall S2M consensus fold, so these may reflect differences in species-specific interactions between S2M and host proteins or RNA molecules.

The deer GWAS also revealed a few nonsynonymous mutations in viral proteins important in RNA binding and host antiviral responses. Specifically, ORF1ab:N4899I (nsp12:N507I) lies in the nsp12 RNA-dependent RNA polymerase, within the highly conserved motif G which is important in positioning the 5′ template strand during viral RNA synthesis (*Sheahan et al., 2020*). While N507 is known to make contact with the +2 nucleotide of the template strand, experimental investigations will be needed to understand how the N507I mutation impacts the active site structure and/or viral RNA synthesis (*Hillen et al., 2020*). The ORF1ab:P5401L (nsp13:P77L) mutation in the nsp13 helicase is predicted to be a solvent-exposed residue within the N-terminal Zinc-binding domain (*Newman et al., 2021*). However, given that this is distant from the helicase active site, we predict that it is unlikely to affect helicase enzymatic activity. Finally, we identified two mutations that may have implications for host adaptation as they are identified in proteins known to interact with the host antiviral response. Specifically, ORF1a:P1640L (nsp3:P822L) in the nsp3 protease localizes to the deubiquitinating site

in the PLpro domain which overlaps with the ISG15 binding site, suggesting it may modulate host antiviral responses (*Yang and Leibowitz, 2015*; *Shin et al., 2020*; *Liu et al., 2021*). Interestingly, the deer GWAS also revealed the ORF7a:T120I mutation within the ORF7a accessory protein. This residue is adjacent to K119, which is implicated in the inhibition of the antiviral response in human cells (*Redondo et al., 2021*; *Cao et al., 2021*). Specifically, K119 polyubiquitination has been shown to block STAT2 phosphorylation, leading to the inhibition of type I IFN. Thus, it is possible that the ORF7a:T120I mutation modulates ubiquitination at the K119 residue; however, this will require experimental validation.

## Concluding perspectives

Upon finalization of this manuscript, another analysis of SARS-CoV-2 transmission and potential host adaptation was published, using a similar dataset from GISAID (*Tan et al., 2022*). Notably, this work focused only on transmission from humans to other animals, while we also considered the animal-to-human direction. The approach used to identify animal-associated mutations was conceptually similar to ours – focusing on homoplasic mutations and a set of reasonable but arbitrary filters for allele frequencies and effect sizes – whereas we took a more formal statistical GWAS approach. Our three mink significant GWAS hits are a subset of the four identified in the other study, and neither study identified any deer-associated mutations in Spike (*Tan et al., 2022*). Our study identified more significantly deer-associated mutations (including the single hit reported by Tan et al. in nsp3, ORF1a:L1853F (nsp3:L1035F)), which could be due to different data filtering and significance testing approaches. Overall, the two studies are complementary and pave the way for future studies on larger datasets.

In conclusion, while the dynamics of anthroponosis and zooanthroponosis for SARS-CoV-2 are still unclear, cross-species transmission events are likely to continue given continued geographically widespread infections, high transmission rates, and the emergence of new variants. We identified several statistically significant animal-associated substitutions in mink and deer, suggestive of sustained animal-to-animal transmission and perhaps reflective of host adaptation. Continuous molecular surveillance of SARS-CoV-2 from animals is likely to reveal new insights into SARS-CoV-2 host range and adaptation and may contribute to our understanding of the risk for spillback of new variants. This also highlights the need to monitor for similar patterns of susceptibility to infection and sustained intra-species transmission in related species. Our study draws attention to several specific, statistically significant nucleotide and amino acid substitutions that may play a role in host adaptation, pathogenesis, and/or transmission and are candidates for experimental study.

## Methods
### Data

On February 28, 2022, we downloaded from GISAID (*Shu and McCauley, 2017*) all SARS-CoV-2 consensus genome sequences derived from non-human animals. These host species include several mammalian species such as whitetail deer, mink, cats, dogs, lions, and monkeys among others. The raw non-human dataset obtained from GISAID was filtered for low-quality sequences, sequences with a length of less than 29,000 base pairs, and an ambiguous nucleotide (N) count above 500. Sequences with incomplete dates recorded in the metadata were discarded and excluded from the study. This study focuses on species with at least 50 sequences in the dataset, namely mink (n=1038), deer (n=134), cat (n=78), and dog (n=59).

From the 7.6 million human-derived viral sequences present in GISAID's human alignment dated February 28, 2022, a set of closely related sequences for every animal species was extracted. To do so, we used Nextstrain (*Hadfield et al., 2018*) to calculate a proximity matrix based on pairwise substitutions between every animal-derived sequence and all ~8 million human-derived sequences. The query alignment was generated using MAFFT (*Katoh and Standley, 2013*) in which animal sequences were aligned to the gapped version of the Wuhan IV04 reference sequence present in GISAID's human-host alignment dated February 28, 2022, with the 'keeplength' option of the software enabled to maintain the length of context alignment in the query alignment.

Using this matrix and noting that some sequences from a given species might share a number of close relatives, we extracted the closest human-derived sequences for every sequence of a given species such that the unique set of best-hit human-host sequences for a particular species would have

roughly the same count as the sequences from that animal species. To provide a greater and more representative phylogenetic context, 500 human-derived sequences were subsampled randomly from every month of the pandemic, from January 2020 to February 2022, resulting in 13,000 human-derived sequences, distributed uniformly over time. Here again, the same quality filtering criteria were applied and the sample was drawn from sequences that were at least 29,000 base pairs long and had fewer than 500 ambiguous nucleotides ('N' characters). The GISAID identifiers for all sequences used in this study are reported in *Supplementary file 1*.

## Alignment

All the above sequences, that is, the animal-host sequences along with their closely related human host sequences and the context random human subsample, and the Wuhan IV04 reference sequence were aligned using MAFFT (version 7.471). We used the NW-NS-2 setting, which is a speed-oriented, progressive method without FFT approximation. The alignment was done in two steps; first, a preliminary alignment was done and problematic sequences causing almost invariant insertions due to stretches of ambiguous nucleotides (N's) were removed from the dataset, and the remaining sequences were aligned again with the exact same algorithm, resulting in the final alignment, including a total of 14787 sequences, comprised of 1038 mink sequences, 134 deer sequences, 39 dog sequences, and 78 cat sequences. As for the closely related human-host sequences, this dataset included 852 close relatives for mink, 61 for cat, 31 for dog, and 123 for deer. The remaining sequences were human-host sequences randomly subsampled.

## Phylogeny

Ten replicate trees for each candidate species were generated, each of which included the focal species' sequences, its closely related human-host sequences, and one-tenth of the randomly subsampled human-host sequences which were again split uniformly over time to ensure temporal heterogeneity, resulting in an overall count of 40 trees. Maximum-likelihood divergence trees were inferred using IQ-TREE (*Nguyen et al., 2015*) with a general time-reversible model and with 1000 bootstrap iterations using the ancestral sequence reconstruction (-ASR) option. The resulting trees were pruned for branches that are unreasonably divergent, and tips whose lengths were considered outliers according to the interquartile criterion and were longer than $q_{0.75} + 1.5 IQR$ were pruned (q0.75 = 3rd quartile; IQR = q0.75-q0.25; q0.25=1st quartile). This criterion discarded a maximum of five deer sequences out of 134, a maximum of two mink sequences out of 1038, and a maximum of one dog sequence out of 39, while no cat sequences were pruned.

## Ancestral state reconstruction

Discrete ancestral state reconstruction was performed on all 40 trees in the study, with states set as 'human' and 'animal.' The reconstruction was done using the maximum likelihood method 'ace' implemented in the 'ape' package (*Paradis and Schliep, 2019*) in R (*R Development Core Team, 2021*). Ancestral state reconstruction was done using the 'equal rates model' which allows for an equal probability of transition in both directions of 'animal-to-human' and 'human-to-animal' on the tree.

## Estimating transmission events

Once the states were labeled, the set of all branches along which a transition from an 'animal' node state to a 'human' node state has occurred was identified, and the human end of the branch, namely the most recent common ancestor of the introduced human clade or the human tip in case of single-tons, was marked as a transition node. In order to identify an independent set of transitions and to avoid the redundancy of reporting nested spilled-back clades as separate events, the most basal of these transition nodes was identified. In order to provide a well-rounded record of spillovers, the same analysis was done in the human-to-animal direction as well. For setting a lower bound to the transmission counts, in another set of analyses, all transitions in the desired direction were identified, then the branches whose parent side had a bootstrap support of lower than 75% were discarded, and subsequently, the most basal transitions were identified. To validate the ancestral state reconstruction and the transmission counts, a permutation test was done in which the tips of every 10 trees for all candidate species were randomly shuffled in 1000 permutations, and the same algorithm for ancestral state reconstruction and subsequently counting transitions in both directions (animal-to-human and

human-to-animal) was applied to find the most basal transitions, unfiltered counts on the shuffled trees.

## Genome-wide association studies

We used POUTINE (*Chen and Shapiro, 2021*) to scan the genome for mutations that are statistically associated with each animal species. POUTINE relies on the viral phylogeny to identify homoplasic mutations associated with a phenotype of interest, in this case, human vs. animal hosts. By considering only homoplasic mutations at the tips of the tree, POUTINE implicitly accounts for population structure. POUTINE was run on 40 tree replicates: 10 per species, with each replicate containing the same sequence alignment used in the transmission analysis, though trees were inferred without the -ASR option in IQ-TREE (Ancestral sequence reconstruction was unnecessary because POUTINE only tests mutations that occur on terminal branches). In each run, animal-host sequences were set as 'cases' and human-host sequences as 'controls.' We chose to treat animals as cases because the root of the tree is human, and initial transmission events are human-to-animal with more recent and rare animal-to-human events. The dataset is, therefore, better suited to identify animal-associated mutations than human-associated mutations. In each of the ten replicates for every species, any sites with a minor allele familywise p-value of less than 0.05 were recorded as a hit for that run. The unique collective set of hits for every species across all 10 runs was retained and the number of replicates in which each hit appeared was recorded.

## Fisher's exact test for geographic bias

In order to check for geographic bias we performed a Fisher's exact test on every SNV identified by POUTINE to compare its frequency in human hosts inside and outside regions where animal-host sequences bearing these mutations are found. For each SNV, we obtained its geographical distribution of human-host sequence counts broken down by geographical division from CoV-Spectrum. We partitioned global divisions into regions that animal-host sequences bearing the specific mutation are found ('in' regions), and regions in which such sequences are not found ('out' regions). We then created the 2-by-2 contingency table in which rows correspond to wildtype allele and alternate (animal-associated) allele counts, and columns correspond to 'in' or 'out' region counts. We then calculated the pointwise estimate and confidence interval for the odds ratio, and flagged mutations in which the two frequencies were significantly different. Counts and frequencies for this step are recorded in *Supplementary files 4 and 5*.

## Code availability

Scripts used to perform all analyses are available at https://github.com/Saannah/Animal.SARS-CoV-2.git, (copy archived at *Naderi, 2023*).

## Acknowledgements

We gratefully acknowledge all data contributors, i.e., the Authors and their Originating laboratories responsible for obtaining the specimens, and their Submitting laboratories for generating the genetic sequence and metadata and sharing via the GISAID Initiative, on which this research is based. We are grateful to CoVaRR-Net colleagues Sally Otto, Art Poon, Caroline Colijn, Will Hsiao, Fiona Brinkman, Paul Gordon, Arinjay Banerjee, Jason Kindrachuk, Angie Rasmussen, and Samira Mubareka for useful discussions that helped improve the manuscript. We also thank the two peer reviewers for their constructive comments. This study was supported by the Canadian Institutes for Health Research (CIHR) operating grant to the Coronavirus Variants Rapid Response Network (CoVaRR-Net). Data analyses were enabled by computing and storage resources provided by Compute Canada and Calcul Québec.

## Additional information

### Funding

| Funder | Grant reference number | Author |
|---|---|---|
| Canadian Institutes of Health Research | Coronavirus Variants Rapid Response Network (CoVaRR-Net) | B Jesse Shapiro Selena M Sagan |

The funders had no role in study design, data collection and interpretation, or the decision to submit the work for publication.

### Author contributions

Sana Naderi, Data curation, Software, Formal analysis, Investigation, Methodology, Writing – original draft, Writing – review and editing; Peter E Chen, Software, Methodology; Carmen Lia Murall, Supervision, Methodology; Raphael Poujol, Susanne Kraemer, Data curation, Supervision; Bradley S Pickering, Conceptualization, Writing – review and editing; Selena M Sagan, Conceptualization, Supervision, Funding acquisition, Writing – original draft, Writing – review and editing; B Jesse Shapiro, Conceptualization, Supervision, Funding acquisition, Writing – original draft, Project administration, Writing – review and editing

### Author ORCIDs

Sana Naderi ⬡ http://orcid.org/0009-0009-8310-3657
Selena M Sagan ⬡ http://orcid.org/0000-0002-8484-3632
B Jesse Shapiro ⬡ http://orcid.org/0000-0001-6819-8699

### Decision letter and Author response

Decision letter https://doi.org/10.7554/eLife.83685.sa1
Author response https://doi.org/10.7554/eLife.83685.sa2

## Additional files

### Supplementary files

• Supplementary file 1. GISAID accession numbers of all sequences used in this study.

• Supplementary file 2. Number of viral sequences passing quality filters. The counts show the initial number of sequences downloaded from GISAID from each animal species, and the remaining number after each consecutive quality filter. The 'quality control' count shows the number of sequences after removing those with incomplete sampling dates and/or >500 ambiguous bases (Ns). The 'post-alignment pruning' shows the count after removing sequences shorter than 29,000 bases and/or with an insertion absent in all other sequences (introducing a gap in the alignment). The 'divergent tree branches' shows the count after removing sequences that introduce long branches into the phylogeny (Methods). Ranges of counts indicate variation across tree replicates.

• Supplementary file 3. Table of transmission counts for all candidate species, in both animal-to-human and human-to-animal direction, for both bootstrap-filtered and unfiltered cases.

• Supplementary file 4. Human-derived sequence counts bearing each of the significant GWAS hits identified in deer inside and outside regions where deer sequences containing each mutation are found. Odds ratio and the p-values are reported following a Fisher's exact test. GWAS hits with OR <1 or not significantly different from 1 are highlighted in green.

• Supplementary file 5. Human-derived sequence counts bearing each of the significant GWAS hits identified in mink inside and outside regions where mink sequences containing each mutation are found. Odds ratio and the p-values are reported following a Fisher's exact test. GWAS hits with OR <1 or not significantly different from 1 are highlighted in green.

• Supplementary file 6. Number of times Mink GWAS hits appear along human-to-mink transmission branches. The counts are summed across all branches and all 10 tree replicates.

• Supplementary file 7. Number of times Deer GWAS hits appear along human-to-deer transmission branches. The counts are summed across all branches and all 10 tree replicates. Yellow-colored rows are mutations that never appear on a human-to-animal transmission branch over all deer replicates. The orange-colored row corresponds to a nucleotide position mutated along human-to-animal

transmission branches, but the substitution was never identical to the animal-associated allele identified by GWAS at that position.

• Supplementary file 8. SARS-CoV-2 mutations were previously associated with non-human animal species in the literature. Insertions and deletions are not considered. For each of the studies, mutations reported in the main text or main tables are included. For the (*Pickering et al., 2022*) study, substitutions found in deer in their study and also appear at least once in the database of previously reported deer sequences (listed in the paper's *Supplementary file 2*) are included.

• MDAR checklist

## Data availability

All data used in this study is available at gisaid.org.The GISAID identifiers for all sequences used in this study are reported in Supplementary File S1 and can be found at https://doi.org/10.55876/gis8.230328xz.

The following previously published dataset was used:

| Author(s) | Year | Dataset title | Dataset URL | Database and Identifier |
|---|---|---|---|---|
| McCauley S | 2022 | GISAID | https://doi.org/10.55876/gis8.230328xz | EPI_SET_230328xz, 10.55876/gis8.230328xz |

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
