## [Editor Report]

The authors have rigorously examined the existence of mutations that are associated with SARS-CoV-2 transmission between non-humans and human animals. The findings are important for contemporary conversations around viral niche breadth and disease emergence. The evidence provided is very convincing, and the authors have creatively demonstrated how these sorts of GWA studies can be used to identify the genomic signature of niche breadth.

---

## [Decision Letter]

**Decision letter after peer review:**

Thank you for submitting your article "Zooanthroponotic transmission of SARS-CoV-2 and host-specific viral mutations revealed by genome-wide phylogenetic analysis" for consideration by *eLife*. Your article has been reviewed by 2 peer reviewers, and the evaluation has been overseen by a Reviewing Editor and Detlef Weigel as the Senior Editor. The following individual involved in the review of your submission has agreed to reveal their identity: Nash Rochman (Reviewer #1).

Essential revisions:

1) Please address suggestions made with regard to clarity, methodological details, and other aspects of the structure of the paper, as identified by the reviewers. This will go a long way towards maximizing the impact of this paper.

2) Please address suggestions made by both reviewers with respect to the figures, analyses, and visuals.

*Reviewer #1 (Recommendations for the authors):*

I find this work to be clearly presented and of broad interest. I believe the manuscript is publishable in its current form and that any changes should be made at the authors' discretion; however, I hope the authors will consider addressing the below comments prior to publication.

Principal comments: I agree with the methods used and the statistics present. My only substantive comments are regarding the presentation. First, I encourage the authors to consider adding an additional main figure summarizing key mutations with putative functional relevance, newly proposed in this manuscript; previously proposed and validated in this manuscript, and previously proposed but which do not meet the significance criteria for this analysis. I recommend depicting the full range of non-human hosts from which SARS-CoV-2 isolates have been obtained in this figure as well, even if those hosts were excluded from this study due to data availability.

Second, I encourage the authors to consider modifying the text throughout to clarify what was unknown prior to this work. As written, I believe many readers will not fully appreciate the novelty of this study. While I feel the presentation of the novel identification of mutations with putative functional relevance and validation of previous adaptive mutations is clear, I believe the novelty and value of the comprehensive cataloguing of cross-species transmission events (which I feel is the principal novel result) could be further emphasized. Perhaps the authors would consider expanding the introduction to include a brief statement regarding why "obvious" methods to detect cross-species transmission events from manually or qualitatively reviewing the global SARS-CoV-2 phylogeny are intractable.

Additionally, I think the text would benefit from a brief discussion regarding how this problem was previously addressed for MERS (highly relevant as both a betacoronavirus and a virus with many validated cross-species transmission events, https://elifesciences.org/articles/31257). Perhaps the authors would consider adding a statement to the discussion indicating how this analysis robustly demonstrates the need for increased sequencing efforts of animal populations given the substantial imbalance between the host diversity and likely large number of cross-species transmission events and the modest number of such events which may be inferred from the retrospective analysis.

*Reviewer #2 (Recommendations for the authors):*

I would like to commend and congratulate the authors for this contribution to the literature. In this case, the content and research goals associated with this manuscript are not only of interest to basic science research in evolutionary biology but also have relevant and critical importance to pandemic containment and global human health outcomes. I look forward to following this work through the review process and to publication.

The goals of the paper are clearly spelled out and tested, and there is no concern that there is a mismatch between what was intended for the study by the authors and what was delivered. However, my major critique of the manuscript is that it reads as if the work has been rushed, including the following key points: (1) Discussion of the dataset being unbalanced in terms of available SARS-CoV-2 genomic sequences across species and how that may impact results, but lending no quantitative solution or detailed qualitative justification to that problem, (2) absence of detailed discussion about previously well-explored factors about transmission across species, including phylogenetic and structural similarity of non-human ACE2 to hACE2 and human-wildlife contact, and (3) a series of large and small improvements to main manuscript figures and tables that would make them more informative, relevant, and/or easier to tie to main results. I understand that the SARS-CoV-2 space may be competitive right now and maybe there is a rush to get the work out, but small efforts in these areas I have outlined could help this manuscript make a bigger impact, and I hope that is how you will interpret my advice.

Item 1: Several times throughout the manuscript you mention that the dataset is unbalanced across species because of differences in the sequencing effort of SARS-CoV-2 across non-human species. This is to be expected because non-human taxa have not, as yet, received a lot of attention, lending strength and importance to your efforts here. I think the main thing is how to tackle that without underselling your efforts or making your reader question the validity of your results. You mention that there are better statistical tests for handling datasets constructed like this, but you don't employ them. You do take steps in some of your analyses to employ resampling techniques to compensate for differences in sequencing effort. You never offer a qualitative assessment of why you did/didn't do certain things. You could probably ease this concern for your reader by one of two solutions: (1) do the quantitative analysis that you suggest in the paper would be better for a sample dataset like this, or, (2) the first time you mention the unbalanced dataset, offer a few sentences to justify why the analyses you did were fine, still valuable tests, and how you implemented some mitigation steps along the way. Then, your reader is aware of the limitations or mitigations from the beginning and is not left wondering why you say in the manuscript (Line 255/266) that there was a better test/statistical method that you didn't use.

Item 2: In the non-human literature on SARS-CoV-2 to date, there has been a lot of focus on two central themes regarding zoonotic/reverse-zoonotic transmission and evolution of SARS-CoV-2 in non-human hosts, including the phylogenetic and structural similarity of non-human ACE2 to hACE2 and the degree of human-wildlife contact giving opportunity for spillover/back events. You mention both of these things briefly in places in the introduction and discussion, but never really tie the whole picture together. For example, does ACE2 similarity, either in terms of phylogenetics or molecular structure, make sense with the viral evolution results you have across your small group on non-human taxa? Does this hold up or is it all over the place and less informative than we have been led to believe? You talk about human contact in mink farms being high, but would we expect it to be higher than human contact with their domestic pets? Is it a combination of these two factors that best explain your results? Or something else regarding the epidemiology of SARS-CoV-2 in these species that may be contributing to the different mutation rates and cross-species transmission events. I think one solid paragraph that was truly comparative with all the species you included and the ideas we had prior to this manuscript about what might be important for reverse zoonotic transmission, followed by within non-human species transmission and evolution, would be really helpful and important. You discuss a lot of nuances of specific within-species SNVs, it is worth thinking about whether you can call attention to or hypothesize about any broad patterns associated with host species biology or SARS-CoV-2 epidemiology.

Item 3: Figures 1a-d are ineffective for me. I realize the main point being made is the overall picture of a number of occurrences of mutations and transmissions across non-human species, but these phylogenies could be cleaned up and more readable. For example, you could color just the non-human host branches of each viral phylogeny and leave human host branches uncolored (black) and provide a better description in your figure caption, including details of sample sizes and the dataset in general. Figures 1e/f don't really need the violin plot portion. They are almost unreadable/unrecognizable anyway. You could just use raw data points and log-transform the y-axis for better visualization of the data. These two plots should also be paneled to share the same y-axis and x-axis labels. In Tables 2/3, there is no function to the use of the color here and, therefore, no need for it. In Figure 3, I love the genomic annotation built into the x-axis, but I would not use the same color blue in the annotation as you do in the graph. I would also panel these graphs so that there is only one x-axis and only one y-axis label. You could label species inside each panel.

---

## [Author Response]

Essential revisions:1) Please address suggestions made with regard to clarity, methodological details, and other aspects of the structure of the paper, as identified by the reviewers. This will go a long way towards maximizing the impact of this paper.

We have carefully addressed all the reviewer comments below, including a new Figure 1 summarizing mutations previously associated with different animal species, along with those supported with statistical significance in our GWAS. We have also made several clarifications throughout the text, as suggested. We also note that one of the reviewer comments spurred us to run phylogenetic inference with ancestral sequence reconstruction (ASR), which revealed a bug in the IQ-TREE software. In short, the bootstrap values were not being output in a proper format without the ASR option. We have now updated the inferred human-to-animal and animal-to-human transmission counts (Table 1 and Figure 3) with the correct bootstrap values. While the exact counts have changed somewhat, the results are qualitatively the same and our conclusions remain unchanged.

All computer code for the original and additional analyses are available and documented on the manuscript’s GitHub page, under a folder named *eLife*: https://github.com/Saannah/Animal.SARS-CoV-2

2) Please address suggestions made by both reviewers with respect to the figures, analyses, and visuals.

In addition to the new Figure 1, we have replotted the phylogenies (Figure 2) in a circular orientation with unscaled branch lengths to allow better visual clarity of the inferred cross-species transmission events along the tree topology. We have also replotted the dot plots, removing unnecessary violin plots and improving the color palette (Figure 3). We also adjusted the color palette of the gene annotations along the horizontal axis of the Manhattan plots (Figure 4).

Reviewer #1 (Recommendations for the authors):I find this work to be clearly presented and of broad interest. I believe the manuscript is publishable in its current form and that any changes should be made at the authors' discretion; however, I hope the authors will consider addressing the below comments prior to publication.

We appreciate this assessment and we carefully consider all the comments below.

Principal comments: I agree with the methods used and the statistics present. My only substantive comments are regarding the presentation. First, I encourage the authors to consider adding an additional main figure summarizing key mutations with putative functional relevance, newly proposed in this manuscript; previously proposed and validated in this manuscript, and previously proposed but which do not meet the significance criteria for this analysis. I recommend depicting the full range of non-human hosts from which SARS-CoV-2 isolates have been obtained in this figure as well, even if those hosts were excluded from this study due to data availability.

Thank you for this comment. As suggested, we have created a new Figure 1 showing the full range of non-human animal species from which SARS-CoV-2 genomes have been deposited in the GISAID database. In a second panel, we show a heatmap summarizing mutations previously associated with different animal species, along with those statistically supported in our GWAS. We note that this heatmap includes key prior studies, but is not a comprehensive meta-analysis (which would be beyond the scope of our study).

Second, I encourage the authors to consider modifying the text throughout to clarify what was unknown prior to this work. As written, I believe many readers will not fully appreciate the novelty of this study. While I feel the presentation of the novel identification of mutations with putative functional relevance and validation of previous adaptive mutations is clear, I believe the novelty and value of the comprehensive cataloguing of cross-species transmission events (which I feel is the principal novel result) could be further emphasized. Perhaps the authors would consider expanding the introduction to include a brief statement regarding why "obvious" methods to detect cross-species transmission events from manually or qualitatively reviewing the global SARS-CoV-2 phylogeny are intractable.

Thanks for this suggestion. We have expanded the following paragraph in the Introduction to further emphasize the benefits of our approach:

“While such case studies have been valuable in highlighting patterns of transmission and adaptation across individual animal species since the beginning of the pandemic, a standardized, global analysis of the available data is lacking. In this study, we comprehensively compared SARS-CoV-2 sequences derived from animal hosts using a publicly available dataset. Using phylogenetic methods, we inferred transmission events between humans and four other well-sampled animals and quantified their relative frequencies. Compared to earlier studies which tended to focus on one species at a time and much smaller datasets we use an automated approach to avoid manual browsing of the tree. This not only increases the efficiency of the analysis but also reduces the potential for human error. Secondly, rather than simply identifying all branches involving a transition from one host species to another, our method scans the tree for such branches and retains only the most basal nodes (closest to the root) on the descendant end of a given branch. This approach overcomes the possible overcounting of nested clades as separate transmission events and is thus conservative in reporting possible cross-species transmission events.”

Additionally, I think the text would benefit from a brief discussion regarding how this problem was previously addressed for MERS (highly relevant as both a betacoronavirus and a virus with many validated cross-species transmission events, https://elifesciences.org/articles/31257). Perhaps the authors would consider adding a statement to the discussion indicating how this analysis robustly demonstrates the need for increased sequencing efforts of animal populations given the substantial imbalance between the host diversity and likely large number of cross-species transmission events and the modest number of such events which may be inferred from the retrospective analysis.

Thanks for this suggestion. We have now cited the MERS paper in the Introduction as follows:

“Viral adaptation to one host species might result in the tradeoff of reduced transmission in other species. For example, the Middle East respiratory syndrome coronavirus appears primarily adapted to camels, with relatively short-lived transmission chains in humans (Dudas et al. 2018). Alternatively, evolution in different species could potentially create new paths to peaks in the human-adaptive fitness landscape.”

We also added the following sentence to the end of the first Discussion paragraph, as a clear call for additional sampling:

“As genome sequences continue to be sampled from different animal species, approaches described here will allow researchers to further study the evolution and transmission of SARS-CoV-2 and other pathogens across animal species. Sampling more sequences from diverse animals (including the four analyzed here, as well as others) will enable more robust estimates of secondary zoonotic and zooanthroponotic transmission events and species-specific adaptations.”

Reviewer #2 (Recommendations for the authors):I would like to commend and congratulate the authors for this contribution to the literature. In this case, the content and research goals associated with this manuscript are not only of interest to basic science research in evolutionary biology but also have relevant and critical importance to pandemic containment and global human health outcomes. I look forward to following this work through the review process and to publication.

We appreciate these kind words.

The goals of the paper are clearly spelled out and tested, and there is no concern that there is a mismatch between what was intended for the study by the authors and what was delivered. However, my major critique of the manuscript is that it reads as if the work has been rushed, including the following key points: (1) Discussion of the dataset being unbalanced in terms of available SARS-CoV-2 genomic sequences across species and how that may impact results, but lending no quantitative solution or detailed qualitative justification to that problem, (2) absence of detailed discussion about previously well-explored factors about transmission across species, including phylogenetic and structural similarity of non-human ACE2 to hACE2 and human-wildlife contact, and (3) a series of large and small improvements to main manuscript figures and tables that would make them more informative, relevant, and/or easier to tie to main results. I understand that the SARS-CoV-2 space may be competitive right now and maybe there is a rush to get the work out, but small efforts in these areas I have outlined could help this manuscript make a bigger impact, and I hope that is how you will interpret my advice.

This advice is well-taken, and we consider the three points carefully below.

Item 1: Several times throughout the manuscript you mention that the dataset is unbalanced across species because of differences in the sequencing effort of SARS-CoV-2 across non-human species. This is to be expected because non-human taxa have not, as yet, received a lot of attention, lending strength and importance to your efforts here. I think the main thing is how to tackle that without underselling your efforts or making your reader question the validity of your results. You mention that there are better statistical tests for handling datasets constructed like this, but you don't employ them. You do take steps in some of your analyses to employ resampling techniques to compensate for differences in sequencing effort. You never offer a qualitative assessment of why you did/didn't do certain things. You could probably ease this concern for your reader by one of two solutions: (1) do the quantitative analysis that you suggest in the paper would be better for a sample dataset like this, or, (2) the first time you mention the unbalanced dataset, offer a few sentences to justify why the analyses you did were fine, still valuable tests, and how you implemented some mitigation steps along the way. Then, your reader is aware of the limitations or mitigations from the beginning and is not left wondering why you say in the manuscript (Line 255/266) that there was a better test/statistical method that you didn't use.

Thank you for this comment. We agree that we were being overly cautious with the caveats of our methods, at the risk of underselling some of the results. We therefore opted toward the second solution proposed by the reviewer. We attempted to run BASTA (an implementation of a Bayesian structured coalescent approximation) as a method that should minimize the bias in estimates of transmission rates due to different sample sizes for different animal species. However, this software proved too time-consuming to run on our relatively large dataset. (The original BASTA paper described datasets on the order of hundreds, not thousands, of sequences). Moreover, it is not guaranteed to solve the issue of biased sampling since the method requires its own assumptions (priors) on the population sizes of different species, which are ultimately unknown. We therefore prefer to better justify our existing ancestral state reconstruction, while still offering several caveats. This is now done in the Results section as follows:

“Ancestral state reconstruction can be biased by differential sampling across species. Such bias tends to be more severe close to the root of the tree (De Maio et al. 2015); but, in our case, the root is known to be human (as we excluded more divergent animal outgroups). Most of the inferred transmission events are close to the tips and far from the root (Figure 2), which we expect to minimize (but not entirely eliminate) bias. Our goal is, therefore, not to infer absolute rates of cross-species transmission, but rather to provide a consistent comparative framework for interspecies transmission”

And in the Discussion:

“The results are likely biased by uneven sampling effort across species. Specifically, oversampling of human-derived sequences could bias the ancestral state reconstruction toward human-to-animal rather than animal-to-human transmission. In the future, such biases could be accounted for using a Bayesian structured coalescent approximation (De Maio et al. 2015). However, we do not expect this bias to be severe, for three reasons. First, most of our inferred transmission events are close to the tips and bias towards the most common state appears worst close to the root (De Maio et al. 2015). Second, the number of inferred transmission events involving each animal species is not clearly associated with its sample size. Third, we also used permutations to flag certain inferences (e.g. dog-to-human transmission) as unreliable in our current sample.”

Item 2: In the non-human literature on SARS-CoV-2 to date, there has been a lot of focus on two central themes regarding zoonotic/reverse-zoonotic transmission and evolution of SARS-CoV-2 in non-human hosts, including the phylogenetic and structural similarity of non-human ACE2 to hACE2 and the degree of human-wildlife contact giving opportunity for spillover/back events. You mention both of these things briefly in places in the introduction and discussion, but never really tie the whole picture together. For example, does ACE2 similarity, either in terms of phylogenetics or molecular structure, make sense with the viral evolution results you have across your small group on non-human taxa? Does this hold up or is it all over the place and less informative than we have been led to believe? You talk about human contact in mink farms being high, but would we expect it to be higher than human contact with their domestic pets? Is it a combination of these two factors that best explain your results? Or something else regarding the epidemiology of SARS-CoV-2 in these species that may be contributing to the different mutation rates and cross-species transmission events. I think one solid paragraph that was truly comparative with all the species you included and the ideas we had prior to this manuscript about what might be important for reverse zoonotic transmission, followed by within non-human species transmission and evolution, would be really helpful and important. You discuss a lot of nuances of specific within-species SNVs, it is worth thinking about whether you can call attention to or hypothesize about any broad patterns associated with host species biology or SARS-CoV-2 epidemiology.

Thank you for these suggestions. We have now added a new Discussion paragraph under the sub-heading “What explains variation in transmission and evolution across species?” to assess the relative roles of human-animal contact rates and (phylo)genetic similarity (at key receptor sites like ACE2):

“Methodological and sampling considerations aside, there are two major factors affecting rates of transmission across animal species: structural similarity in viral receptor proteins (or more general phylogenetic similarity) and rates of contact between species. These factors have been used to predict cross-species viral transmission, with habitat overlap playing a dominant role over phylogenetic similarity at least on a global scale (Carlson et al. 2022). On more local scales, contact with pets such as cats and dogs will be high with a large number of pet owners, while contact with minks may be high transiently, or for a subset of the population working on farms. Similarly, contact with deer may be restricted to hunters, or may involve indirect contact with human waste. A recent study compared ACE2 orthologs across mammalian species and classified deer ACE2 as having “high” structural similarity to human ACE2; meanwhile cat similarity was “medium”, dog was “low” and mink species were “very low” (Damas et al. 2020). Clearly, large mink outbreaks have occurred despite a relatively dissimilar ACE2 structure – and the ‘mismatch’ between SARS-CoV-2 spike and mink ACE2 could impose strong selection for mink-adapted spike mutations (discussed below). Meanwhile, if deer ACE2 was already structurally similar to human ACE2, this could explain the lack of deer-associated mutations in spike. While it is tempting to speculate that the many other deer-associated mutations outside of spike could provide deer-specific adaptations that reduce transmission back to humans, it is equally possible that the rarity of deer-to-human transmission is explained by relatively low contact rates. As SARS-CoV-2 continues to evolve and transmit in both deer and humans, ongoing sequencing efforts might allow us to test the relative importance of these hypotheses. In summary, while global-scale predictions of zoonotic risk is possible based on contact rates and phylogenetic similarity, it is more difficult to predict the precise risk for a particular species of interest (e.g. mink vs. deer), which depends on detailed life-history and genetic factors. However, combining estimates of habitat overlap with the genomic analyses described here could enable more fine-grained predictions.”

Item 3: Figures 1a-d are ineffective for me. I realize the main point being made is the overall picture of a number of occurrences of mutations and transmissions across non-human species, but these phylogenies could be cleaned up and more readable. For example, you could color just the non-human host branches of each viral phylogeny and leave human host branches uncolored (black) and provide a better description in your figure caption, including details of sample sizes and the dataset in general. Figures 1e/f don't really need the violin plot portion. They are almost unreadable/unrecognizable anyway. You could just use raw data points and log-transform the y-axis for better visualization of the data. These two plots should also be paneled to share the same y-axis and x-axis labels. In Tables 2/3, there is no function to the use of the color here and, therefore, no need for it. In Figure 3, I love the genomic annotation built into the x-axis, but I would not use the same color blue in the annotation as you do in the graph. I would also panel these graphs so that there is only one x-axis and only one y-axis label. You could label species inside each panel.

Thank you for these comments. As suggested, we have replotted the phylogenies (new Figure 2) to make it easier to see the inferred transmission events by plotting circular trees with branch lengths not scaled by genetic distance. (The scaled trees are still available as supplements). We also removed the violin plots and made the panels larger for the dot plots (now their own separate Figure 3) to enhance clarity. Finally, we removed the uninformative color from the GWAS tables (Tables 2 and 3) and corrected the confusing color scheme for the gene annotations along the x-axis of the Manhattan plots (now Figure 4). We believe the clarity of all these display items is now improved.